# Spatio-temporal development of large scale auroral electrojet currents relative to substorm onsets

Sebastian Käki[1,2], Ari Viljanen[1], Liisa Juusola[1], and Kirsti Kauristie[1]

[1]Finnish Meteorological Institute,Helsinki, Finland
[2]University of Helsinki, Helsinki, Finland

**Correspondence:** Sebastian Käki (sebastian.kaki@fmi.fi)

**Abstract.** During auroral substorms the electric currents flowing in the ionosphere change rapidly and a large amount of energy is dissipated in the auroral ionosphere. An important part of the auroral current system are the auroral electrojets whose profiles can be estimated from magnetic field measurements from Low Earth Orbit satellites. In this paper we combine electrojet data derived from the Swarm satellite mission of European Space Agency with the substorm database derived from the SuperMAG ground magnetometer network data. We organize the electrojet data in relation to the location and time of the onset and obtain statistics for the development of the integrated current and latitudinal location for the auroral electrojets relative to the onset. The major features of the behaviour of the westward electrojet are found to be in accordance with earlier studies of field aligned currents and ground magnetometer observations of substorm time statistics. In addition, we show that after the onset the latitudinal location of the maximum of the westward electrojet determined from Swarm satellite data is mostly located close to the SuperMAG onset latitude in the local time sector of the onset regardless of where the onset happens. We also show that the SuperMAG onset corresponds to a strengthening of the order of 100 kA in the amplitude of the median of the westward integrated current in the Swarm data from 15 minutes before to 15 minutes after the onset.

## 1 Introduction

Ionospheric electric currents give rise to a variety of space weather effects that influence the performance and reliability of space-borne and ground-based technological systems. Problems in ground-based systems occur for instance due to geomagnetically induced currents (GIC) in technological conductor systems such as power grids (Pirjola, 2000, 2002). Substorms are a major source of GIC (Viljanen et al., 2006) because the geoelectric fields and induced currents are linked to rapid changes of the ionospheric currents which are highly variable during substorms. A better understanding of the temporal and spatial structure of the high latitude ionospheric currents during substorms and in particular a better description of their contribution for a given time and location is therefore of great importance not only for advances in fundamental space research but also regarding practical applications.

Rostoker et al. (1980) gave a general definition of a magnetospheric substorm as "a transient process initiated on the nightside of the earth in which a significant amount of energy derived from the solar wind-magnetosphere interaction is deposited in the auroral ionosphere and in the magnetosphere" and more specifically as a time interval where most of the energy dissipation

is confined to the auroral oval. Following the definition by Rostoker et al. (1980) the onset of the substorm is associated with a large increase of auroral luminosity in the midnight sector of the auroral oval. The development of the aurora at the onset time and during substorms was first described by Akasofu (1964) who determined that despite the variability from substorm to substorm there are also common features, such as the formation and expansion of the bulge poleward, westward and eastward. Another prevalent phenomenon linked to substorms is the formation of the substorm current wedge (SCW). Bonnevier et al. (1970); Horning et al. (1974); McPherron et al. (1973) established that the SCW is an integral part of substorm physics. The magnetic field signature of the enhanced currents related to the SCW can be observed from ground and the signature also provides a way of identifying substorms and substorm onsets in principle without direct observations of the aurora (Newell and Gjerloev, 2011a; Forsyth et al., 2015). The SCW has been and is still an active research topic, see Kepko et al. (2015) for a review. Ohtani et al. (2021b) found evidence that the nightside subauroral magnetic signatures of substorms can be attributed to the SCW. Recently also the open question of the possible role of small scale wedgelets in forming the large scale SCW has gathered attention (Liu et al., 2015; Nishimura et al., 2020; Ohtani and Gjerloev, 2020; Orr et al., 2021).

The statistical behaviour of the aurora, the enhanced field aligned currents (FAC) linked to the aurora and the SCW as well as the horizontal ionospheric currents related to the SCW have been studied extensively. Gjerloev et al. (2007) used satellite observations in ultraviolet to perform a statistical study of the auroral features described by Akasofu (1964) and obtained a quantitative description of the development of the bulge and the oval aurora. Ohtani et al. (2021a) described the observations of double auroral bulges, Forsyth et al. (2018) studied the seasonal variation of FACs related to substorms from AMPERE data and Coxon et al. (2014) also used AMPERE data to derive statistics of Region 1 and Region 2 FACs during substorms in relation to open magnetic flux. Gjerloev and Hoffman (2014) provided an empirical model of the equivalent current system at the peak of a bulge-type auroral substorm and Orr et al. (2019) used a directed network analysis to estimate the evolution of the equivalent current pattern during substorms. In this study we will use the divergence free current calculated with spherical elementary currents system (SECS) method (Vanhamäki et al., 2003; Vanhamäki and Juusola, 2020) provided by the Auroral Electrojet and auroral Boundaries estimated from Swarm observations (Swarm-AEBS) data products of European Space Agency's Swarm mission (Friis-Christensen et al., 2006). We combine the Swarm-AEBS data set with a SuperMAG substorm list (Gjerloev, 2012; Newell and Gjerloev, 2011b, a) to derive statistics for the divergence free current linked to the auroral electrojets in relation to the substorm onsets. Statistics of the ionospheric currents using the SECS method and Swarm have been derived in previous studies from the viewpoint of hemispheric and seasonal differences (Workayehu et al., 2019, 2020). Using the Swarm data in the substorm context provided by SuperMAG will enable this study to focus on the substorm time divergence free currents. Swarm also provides a different view of the currents compared to ground-based magnetometers as the latitudinal coverage of the auroral oval crossing is not dependent on the network density and the effect of ground induced currents to the magnetometer measurements, which can sum up to tens of percents of the total field strength at ground level (Juusola et al., 2020), is subdued at Swarm altitudes.

In general, the horizontal ionospheric currents can be modelled as sheet currents on a spherical surface with a radius of RE +110 km (Earth radius RE = 6371.2 km). In this thin shell approximation (Untiedt and Baumjohann, 1993) the horizontal ionospheric sheet current density can be separated into two components: the curl-free part, connected to the FACs such that it

closes the regions of upward and downward current, and the divergence free part, forming a rotational current that closes within the ionospheric current sheet (Amm and Viljanen, 1999). The eastward electrojet (EEJ) in the dusk sector and the westward electrojet (WEJ) in the dawn sector are major features associated with the divergence free system. These currents can be studied by using the magnetic field observations from ground and space. Ground-based networks usually provide better spatial coverage and are able to separate spatial and temporal changes in the magnetic field, but the networks are relatively sparse and

can only provide knowledge of the equivalent current pattern which corresponds to the divergence free current. Observations made by satellites and satellite constellations, such as Swarm, CHAMP (CHAllenging Minisatellite Payload) (Reigber et al., 2002) and AMPERE (Anderson et al., 2000, 2002, 2014, 2018; Waters et al., 2001, 2004, 2020; Coxon et al., 2018) which orbit the Earth above the current sheet, can also provide observations about the FAC and the curl free current system along the orbits, but the spatial coverage is usually more limited and it can be difficult to separate spatial and temporal changes. Satellites

on Low Earth Orbit are still relatively close ( i.e. at distances less than 500 km) to the ionospheric currents which enables them to provide information about the ionospheric current system in reasonable latitudinal resolution compared to the auroral oval extent. Signals from structures smaller than the distance between the satellite and ionosphere get strongly attenuated as the magnetic field signature of divergence free currents obey the Laplace equation (Amm and Viljanen, 1999). In particular we can characterize the development of substorm time statistics of the dominant features of the horizontal divergence free currents and

the auroral electrojets using Swarm data. The analysis is done for both the EEJ and the WEJ and we obtain spatio-temporal statistics of the divergence free current carried by auroral electrojets and their boundaries in relation to substorm onset time and location. The structure of the paper is as follows: the data and methods used are described in Section 2 and the results are presented in Section 3. Section 4 contains discussion and Section 5 summarizes the conclusions.

## 2 Data and methods

### 2.1 Satellite and ground-based data

Swarm is a three satellite mission of the European Space Agency to study Earth's magnetic field (Friis-Christensen et al., 2006). Two of the satellites (Alpha and Charlie) were launched to fly side by side with an initial orbital height of 430 km and the third (Bravo) with an orbital height of 530 km. The Swarm-AEBS product is based on the measurements of the Vector Field Magnetometer (Jørgensen et al., 2008). We use the Swarm-AEBS product data for the northern hemisphere and for

Swarm Alpha and Bravo. The data from Charlie results in almost identical data with Alpha and using both Alpha and Charlie would most likely skew the statistics. Swarm-AEBS data contains the electrojet current density and boundaries derived with both the SECS method and the line current (LC) method (Olsen, 1996) and also estimations of the oval boundaries from FAC (Xiong et al., 2014). In this study we use only the SECS based data to determine the integrated currents for auroral oval crossings and the locations of the maxima and equatorward and poleward borders of the electrojets. The current densities have

been derived with the one-dimensional (1-D) SECS method (Vanhamäki et al., 2003; Juusola et al., 2006). The 1-D SECS method is used to determine latitude profiles of the divergence free, curl free, and FAC density for each crossing of the auroral region. The electrojets are defined from the divergence free part of the current. The analysis is performed in an orthogonal

spherical coordinate system (Semi QD) whose pole is rotated to match the pole in the proper Quasi-Dipole (QD) coordinates (Richmond, 1995; Emmert et al., 2010), which are very useful in organising data but do not provide an orthonormal basis for the analysis. In this setup the divergence free current is orientated zonally in the Semi QD coordinate system. However, when we bin the location of the electrojets, we use the QD latitude of the points in question. The electrojets are identified by locating the sign changes in the current density for each oval crossing profile. The latitude values of the sign changes are then used to integrate the profile in the meridional direction in the Semi QD coordinate system between two consecutive sign changes. The operation is thus a line integral along a meridian on the surface of a sphere with a radius of 6481 km, which is the distance from the center of the Earth where the currents are assumed to flow. The WEJ limits are then defined as the coordinates of the sign changes between which the integrated value is most negative and the EEJ limits as the coordinates of the sign changes between which the integrated value is most positive. The peak value is then searched for within both the EEJ and WEJ limits separately. For more details of the detection method we refer to Kervalishvili et al. (2020). Figure 1 shows an example of the divergence free current density for an auroral oval crossing with the detected electrojets. The figure also shows other areas of current in addition to the main electrojets. As mentioned earlier, only the sequences with the most positive and most negative integrated current values are defined as electrojets in the context of this study. Figure 1 also demonstrates that even though the current values are sampled to match the 1 Hz magnetic field measurements used as input data, the 1 degree SECS pole separation provides the scale limit for features in the current.

The SuperMAG substorm list (Gjerloev, 2012; Newell and Gjerloev, 2011b, a) is based on measurements of the SuperMAG ground magnetometer network (Gjerloev, 2012). The list is based on the SuperMAG AL index (SML), which is an auroral electrojet index derived from the SuperMAG data. It is similar to the AL index (Davis and Sugiura, 1966) with the biggest difference being the much greater number of stations used. The latitude and magnetic local time (MLT) coverage of the stations enable the identification of the time, MLT and latitude of the onsets without visual data. Another advantage of SuperMAG based substorm identification is that data availability matches well with the lifetime of Swarm. However, even though the number of SuperMAG stations is large, ocean areas are obviously not well covered and naturally the onset location is dependent on the position of the contributing magnetometers. The onsets in the substorm list have been shown to be highly correlated with a rise in auroral power (Gjerloev, 2012; Newell and Gjerloev, 2011b, a) and the list gives us a temporal relation between the currents and oval boundaries from Swarm measurements and onset parameters (see Fig. 1 and 2). We note that substorms derived using the SOPHIE method could also be used (Forsyth et al., 2015).

## 2.2 Identifying relevant auroral electrojet parameters and isolated substorm onsets

The QD latitude and MLT were calculated for the SuperMAG substorm onsets to match the data with the Swarm-AEBS data. From the onset list we selected all substorm onsets which were more than 2.5 hours apart from the previous one and in the MLT sector between 18–6 hours including the midnight. The 2.5 hour limit is close to what Freeman and Morley (2004) obtained for the periodicity of substorms under constant solar wind driving in their minimal substorm model. We believe this selection gives us the possibility to interpret the times before these onsets as a quieter baseline compared to the times near and after the onsets. Apart from this definition of isolation we do not have any categorization for different scenarios for substorm occurrence

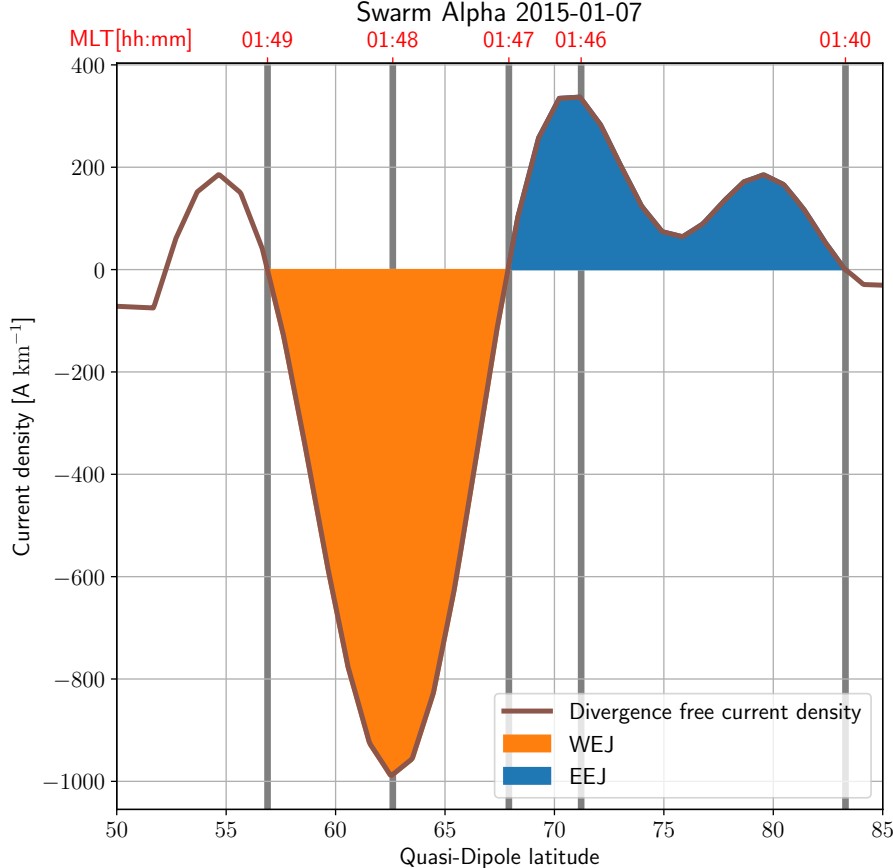

**Figure 1.** An example of the divergence free current derived from Swarm Alpha data with the 1-D SECS method and the identified electrojets from the Swarm-AEBS data set. The location of the boundaries and maxima have been marked with vertical lines. The colored sections show the area corresponding to the integrated currents.

i.e. globally quiet or disturbed magnetospheric conditions or verification of the type of expansion by visible auroras. Figure 2 shows an example of a time series of SML index and the related substorm onsets in relation to the oval crossing in Fig. 1. We do not distinguish cases where there are no recurrent onsets after the initial one from onsets which are followed by recurrent

activity.

In order to relate the onset parameters to the Swarm-AEBS data, we associate timestamps and MLT values for the integrated current values and latitudinal extents for each auroral oval crossing in the Swarm-AEBS data. To do this we use the mean time and MLT sector of the observations which cover the detected electrojets. Because the satellites can cover a large MLT sector close to the poles, the parameters were determined separately for the WEJ and the EEJ. The MLT range covered by a single

oval crossing can exceed 2 hours in some cases. We have used all oval crossings for the time period of 25 November 2013 – 31 December 2019 in the northern hemisphere where both EEJ and WEJ are identified well, i.e. corresponding to the best

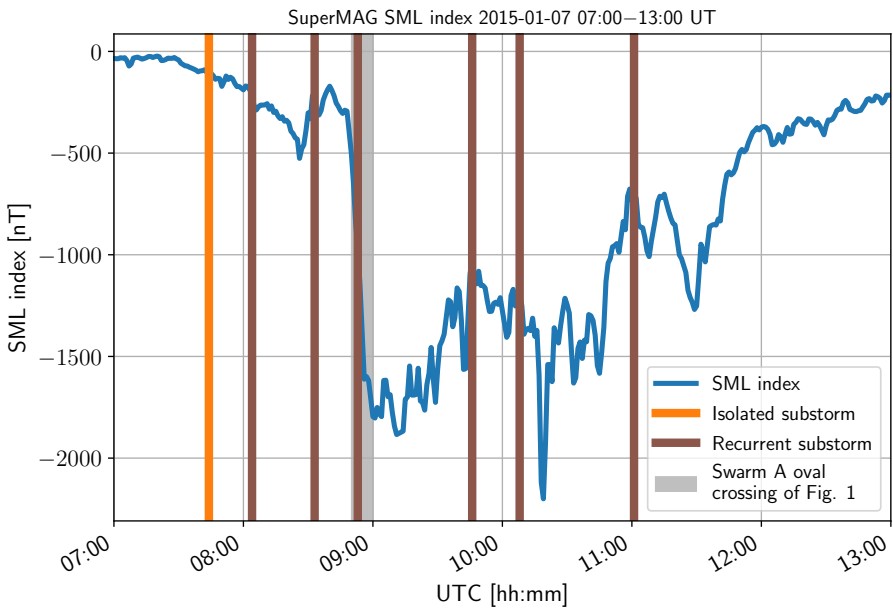

**Figure 2.** An example timeseries of the SML index with the isolated and recurrent substorm onsets marked with vertical lines. The grey shading shows the relation of the oval crossing of Fig. 1 to the SML timeseries.

possible quality flag (Kervalishvili et al., 2020). In practice this means that both the boundaries and the peaks of the electrojets are well defined between the expected auroral electrojet latitude range from 50 to 85 QD latitude and that the satellite path covers the QD latitudes from 50 to 85. Altogether the statistics are calculated from roughly 8430 oval crossings which fulfill

our selection criteria. The crossings cover 2976 onsets from the SuperMAG onset list. For each onset we calculated the MLT and QD latitude so that we can compare the oval crossing parameters to the onsets. The mean MLT and QD latitude of the onsets were 0.15 decimal hours and 67.3 degrees respectively. The integrated WEJ and EEJ values as well as the locations of the maxima and latitudinal extents (see Fig. 1) were then binned in 2 hour bins in MLT difference from the onset MLT and 15 minute bins with respect to the time difference to the relevant substorm onset. The evolution of the parameters of interest are

then inferred from the median and percentiles in each bin. We also further separate the pre-midnight and post-midnight onsets to study the dependence of the data on the MLT of the onset around the onset time.

The binning was chosen to be reasonable with the fact that the SECS method assumes time stationary conditions for the duration of the oval crossing (about 10 minutes) and that divergence free currents are calculated from measurements of the whole oval crossing. We acknowledge that this limits the interpretation of the results to these specific scales. In doing this we

also assume that the integrated currents and the averaged timestamp and MLT sector assigned to it are consistent with each other. Figure 3 shows the number of data points in each bin.

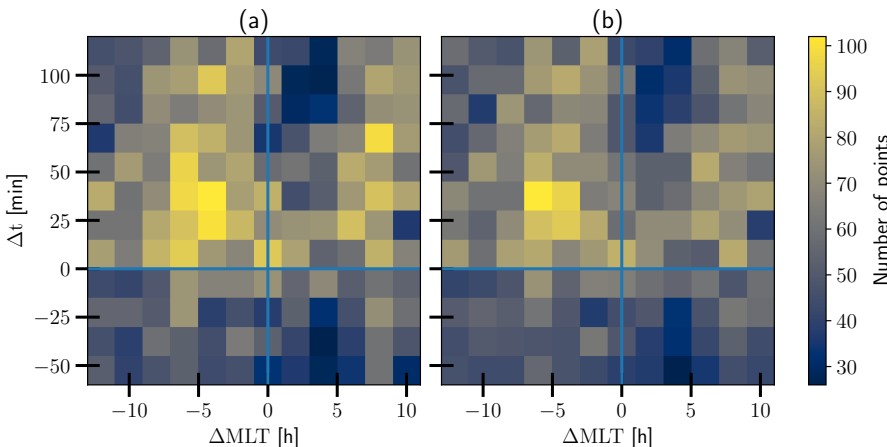

**Figure 3.** The number of Swarm oval crossings in each bin for WEJ (a) (corresponding to panels (a) and (c) in Fig. 4) and EEJ (b) (corresponding to panels (b) and (d) in Fig. 4). $\Delta$MLT is the magnetic local time distance to the onset and $\Delta$t is the temporal distance to the onset.

## 3 Results

### 3.1 General development of the median integrated currents

Figure 4 shows the general development of median integrated WEJ in panel (a) and EEJ in panel (b) with respect to MLT
difference and time offset to the substorm onset. Panels (c) and (d) show the ratio of each median compared to the last median before the onset of the same $\Delta$MLT bin, panel (c) for WEJ and panel (d) for EEJ. For the sake of clearer presentation we have highlighted two MLT sectors in both panels, W1 and W2 in panels (a) and (c) for WEJ and E1 and E2 in panels (b) and (d) for EEJ. Sector W1 stands for 1 hour west (towards smaller $\Delta$MLT values) to 5 hours east (towards larger $\Delta$MLT values) of the onset, sector W2 for 1 to 5 hours west, sector E1 for 11 to 3 hours west and sector E2 for 3 hours west to 3 hours east.
Panels (a) and (b) in Fig. 4 show that the binning organizes the WEJ data better than the EEJ data. As the substorm onset MLT locations in the SuperMAG list are focused heavily around the nightside, we observe traces of the dawn and dusk electrojets in sectors W1 and E1 respectively before the onset, i.e. the negative time difference portion of the plot. A strengthening in amplitude of the WEJ median (corresponding to more negative values) after the onset is clearly visible in sectors W1 and W2. The maximum absolute values of the WEJ are observed 30 to 90 minutes after onset. The medians reach values of about 2 to
3 times the values before the onset and the absolute values of the integrated current in sector W1 can be seen to be about 2 to 2.5 times greater in amplitude than the values in sector W2. The greatest relative increase of the eastward current in panels (b) and (d) can be seen in sector E2 after the onset but the current also increases in sector E1. The intensification in E2 seems to reach the maximum eastward extent only after 15 to 30 min after the onset and the values mostly reach 2 to 2.5 times the values before the onset. We will return to the difficulty in interpreting the EEJ results is Sect. 4.3.

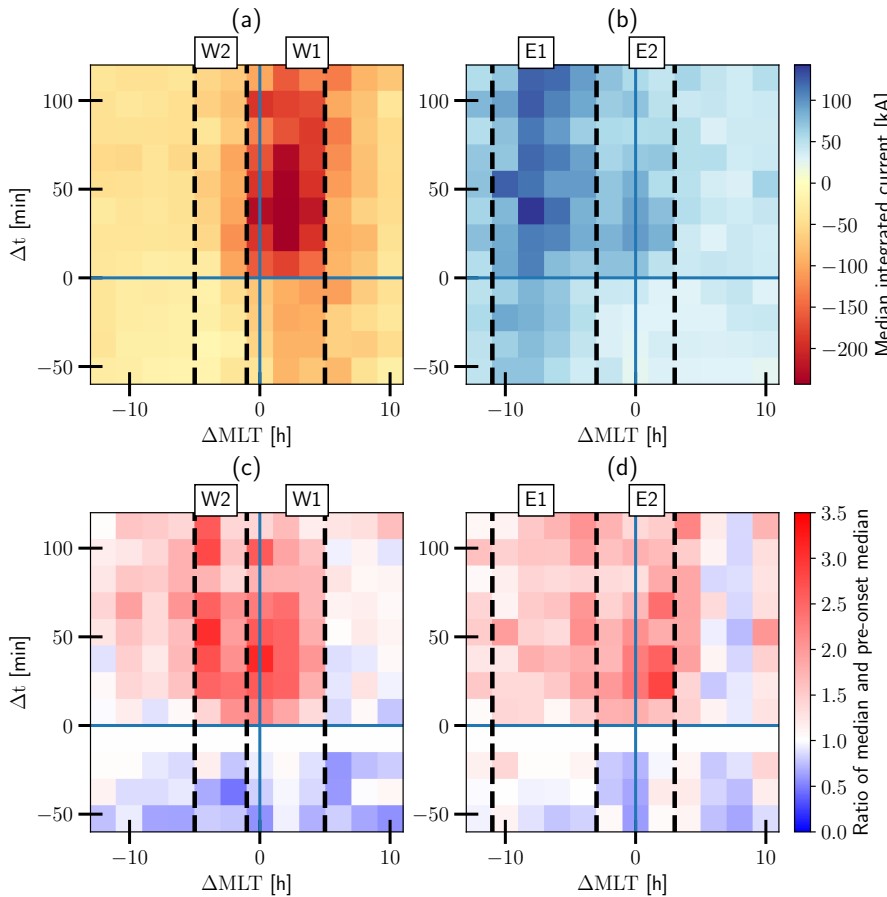

**Figure 4.** The development of the median integrated westward (a) and eastward (b) current binned with 15 minute bins with the respect to the time difference to the substorm onset and 2 hour bin with respect to the MLT difference to the onset. Panels (c) and (d) show the ratio of the medians in panels (a) and (b) compared to the value just before the onset. The dashed vertical lines show the extent of sectors W1, W2, E1 and E2.

## 3.2 Statistics of WEJ and EEJ integrated currents and latitudinal extent

In order to have a more robust view of the binned electrojet currents, we also present figures of medians and the ranges containing the second and third quartiles (when we talk of the range from here onward we mean specifically the range defined like this) of the data overlapped with the plots of the previous time step for the period of 30 minutes before the onset to 75 minutes after the onset in Fig. 5 and 6. In panel (a) of Fig. 5 the distributions of the consecutive time steps are very similar. Comparing the last and first time bins before and after the onset in panel (b), we observe a clear increase of approximately $50 - 150$ kA in the magnitude of the WEJ median and both the upper and lower quartiles mostly in the sectors W1 and W2. Panel (c) shows that from 15 to 30 minutes after the onset the median continues to strengthen in the eastern part of sector W1,

i.e. 1 to 3 hours east of the onset, but the effect is wider in the lower quartile extending completely through both sectors W1 and W2. After 30 minutes there is a well defined sector of strong westward current in the sector W1 with the integrated current median values reaching between -200 and -250 kA. The median values and ranges in sector W2 never drop below -125 kA. Onward from 30 minutes after the onset (panels (d), (e) and (f)) there is very little change in the medians but the quartiles show the large variability of the data with the lower quartile reaching values of roughly -360 kA. The statistics of the EEJ in Fig. 6 show no clearly interpretable development of the distribution in panel (a). Panel (b) shows the development of a second maximum of the EEJ near the onset location in sector E2 in addition to the initial peak in sector E1 which is formed dominantly from the signature of the dusk side EEJ. The magnitude of this peak is rather small as the median reaches only 75 kA. This double peak structure in the median persists in panel (c), but the peak in sector E2 is more concentrated around the middle of the sector as the integrated values rise mostly in the middle and eastern parts of the sector. In panels (d), (e) and (f) of Fig. 6 the double peak structure is still present but the level of large scale organization seems to be decreasing with positive and negative changes both in the medians and ranges in multiple $\Delta$MLT sectors. The largest EEJ values are located in sector E1 with the upper quartile reaching maximum values of little over 200 kA.

Figures 7 and 8 show the medians and ranges for the location of the electrojet with respect to the onset latitude. The shape of the WEJ latitudinal extent as a function of $\Delta$MLT in Fig. 7 is unsurprisingly reminiscent of the shape of the auroral oval of westward flowing current. The shape of the oval of the eastward current can be seen west of the onset in Fig. 8. Looking at all the panels in Fig. 7 we see that the location of the WEJ in sector W1 is quite well centered near the onset latitude at the onset location and around 1 degree poleward of the onset location after the onset time. Panels (c), (d) and (e) of Fig. 7 show that the peak currents seen in the sector W2 are located approximately 2 to 4 degrees poleward of the onset sector currents whereas the W1 sector currents are located consistently closer to the onset latitude. The equatorward and poleward extent indicate a poleward movement of the order of 1 to 2 degrees in the WEJ position after the onset just west of the onset in sector W2. However, keeping in mind that the SECS method resolution is at most one degree to either direction there is uncertainty in the significance of the observation. Fig. 8 panel (a) shows how the EEJ in sector E1 is located poleward of the onset latitude but is clearly located more equatorward close to the onset location. It's also evident that the enhanced EEJ values in the eastern part of sector E2 in Fig. 6 are mostly located poleward of the onset location and WEJ. In sector E2 the median values of the maximum and poleward and equatorward extents of the EEJ location move sharply around 5 degrees poleward at the western end of the sector after the onset in panel (b). However, the medians are located 5 degrees equatorward of the pre-onset position in the extreme western edge of the sector in panels (c), (d), (e) and (f). While the ranges reveal that the data in these panels include EEJ structures similar to the medians in panel (b) and vice versa, the sharp jump in the median location persists but is now located in the middle of sector E2. This sharp edge of over 5 degrees is in contrast to the smoother poleward transition of the medians of panel (a). This is most likely caused by the enhanced WEJ dominating the E2 sector so that the EEJ is found either poleward or equatorward of the WEJ which naturally leads to this splitting of the distribution into two populations.

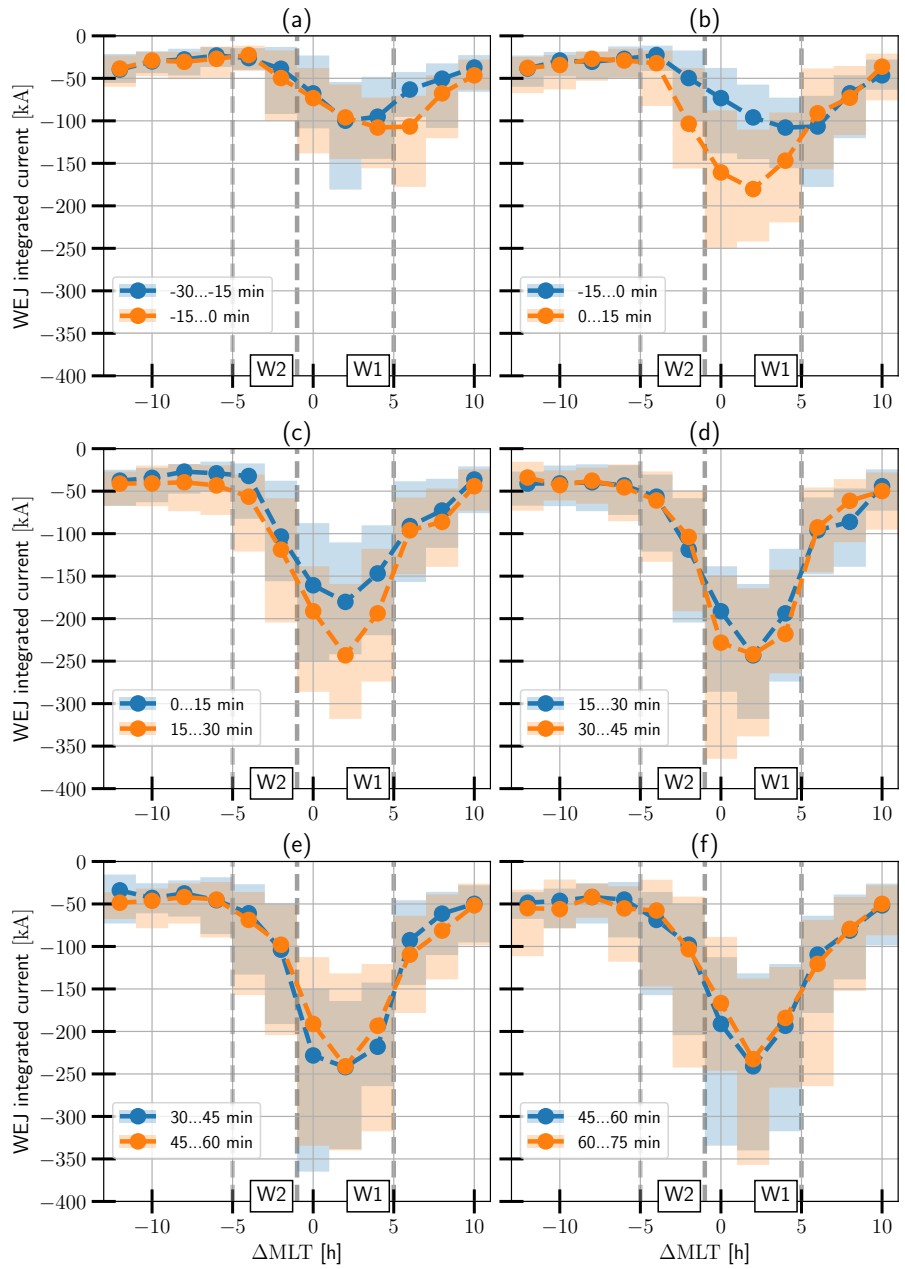

**Figure 5.** The evolution of the WEJ compared to the previous time step for the time period of 30 minutes before to 75 minutes after the onset. The lines show the medians and 50 % of the values are contained within the bars in each bin.

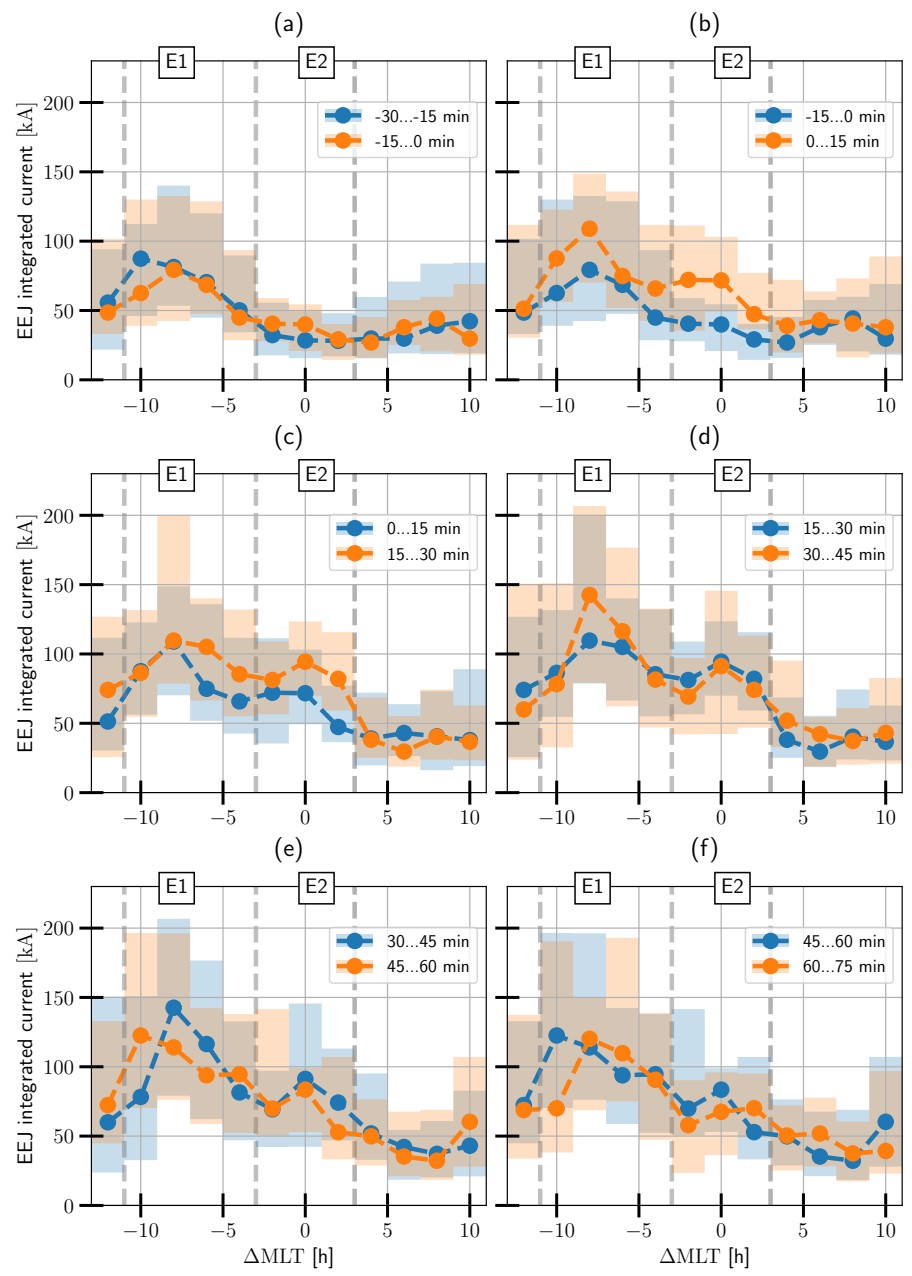

**Figure 6.** The evolution of the EEJ compared to the previous time step for the time period of 30 minutes before to 75 minutes after the onset. The lines show the medians and 50 % of the values are contained within the bars in each bin.

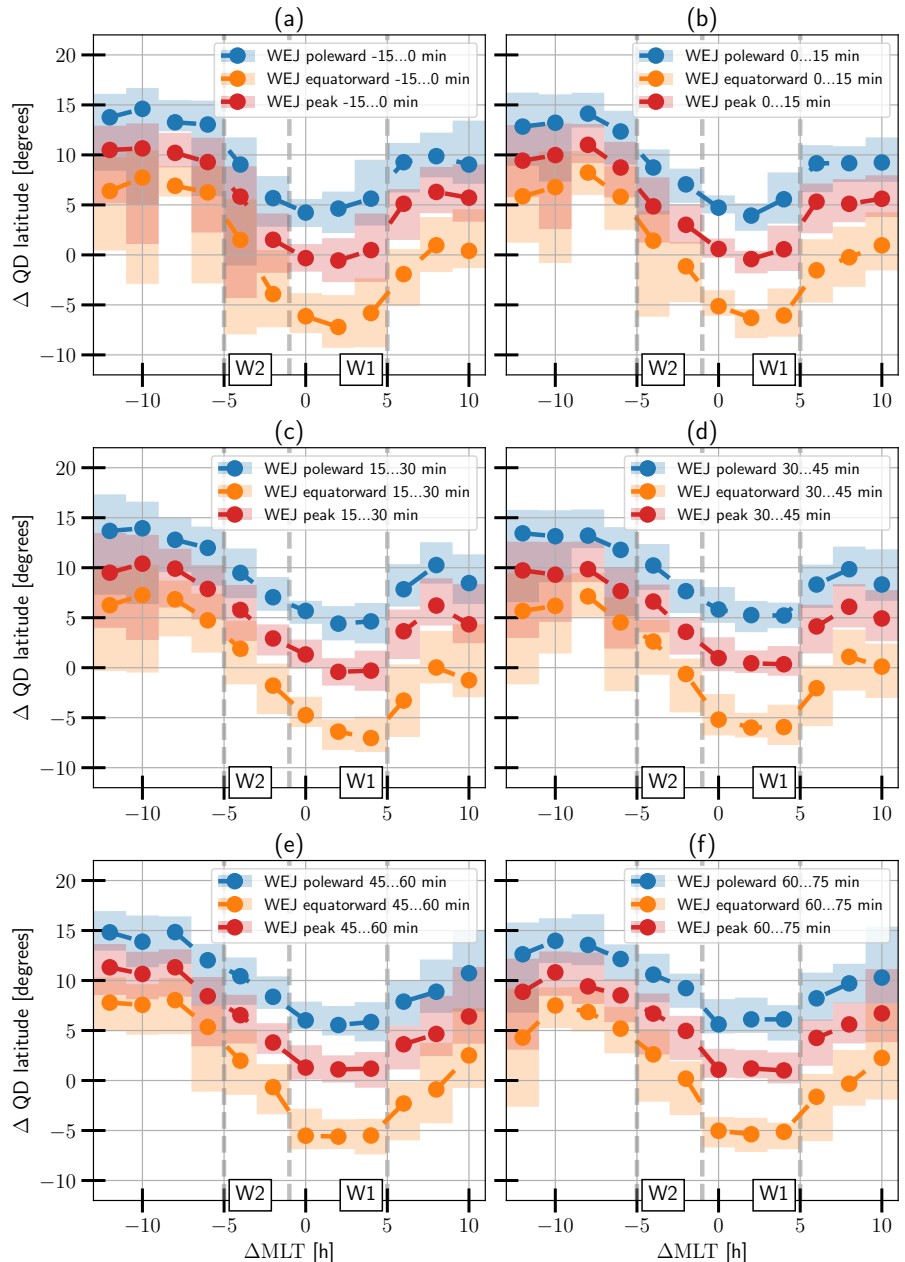

**Figure 7.** The evolution of the WEJ peak location and extent for the time period of 15 minutes before to 75 minutes after the onset. ΔQD is the latitude relative to the onset latitude. The lines show the median and the shadings show the range covered by the second and third quartiles.

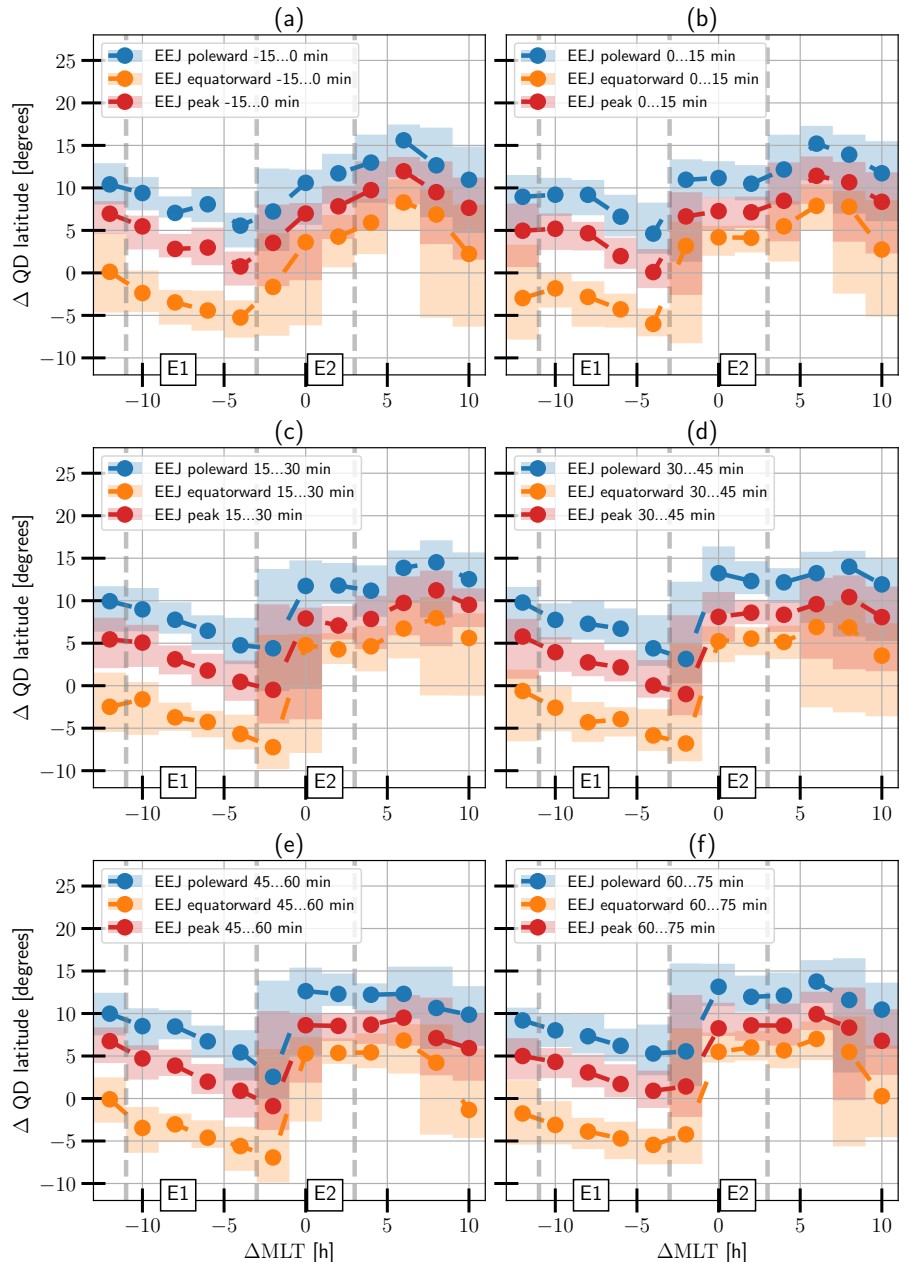

**Figure 8.** The evolution of the EEJ maxima location and extent for the time period of 15 minutes before to 75 minutes after the onset. $\Delta$QD is the latitude relative to the onset latitude. The lines show the median and the shadings show the range covered by the second and third quartiles.

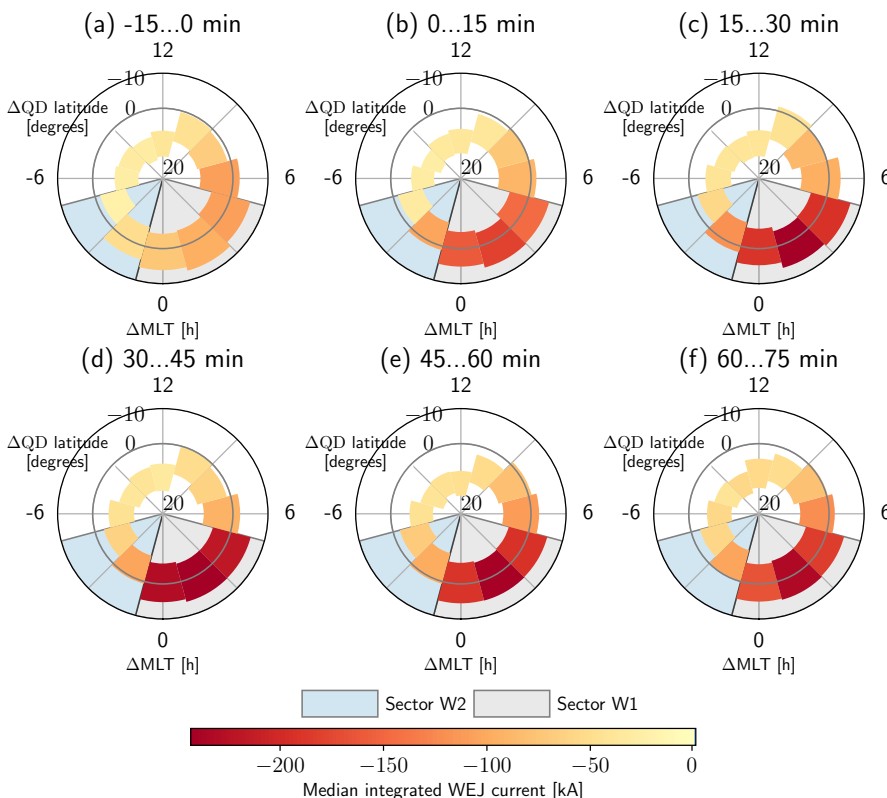

**Figure 9.** The time evolution of the WEJ median current and extent of the jet in polar $(\Delta QD, \Delta MLT)$ coordinates.

To sum up the median behaviour of the WEJ data we present the combined time development of current and the location of the WEJ in Fig. 9 illustrating that the W1 sector currents are greater than W2 currents and the jet in sector W1 is positioned around the onset location in contrast to the poleward position of the jet in sector W2. The apparent poleward displacement of the median WEJ location after the onset in the eastern edge of sector W2 is also visible.

### 3.3 Dependency of WEJ parameters and evolution on the onset MLT

Figure 10 shows the WEJ latitude location and MLT data from the 2 hour MLT bin centered on the onset MLT, i.e. the $\Delta MLT = 0$ bin, and the time step corresponding to the period between 0 and 15 minutes after the onset. We see that although the onset MLT distribution is spread out, the point distribution is consistently such that the $\Delta QD = 0$ is close to the peak values and between the equatorward and poleward borders for onsets between 21–06 MLT. However, for onsets between 18–21 MLT the peak location moves poleward of the onset MLT and the furthest duskward jets are located nearly completely poleward of the onset, although the number of cases is low in this sector. It is possible that the poleward displacement is a signature arising from the Harang discontinuity (Koskinen and Pulkkinen, 1995). The figure also shows that the latitudinal extent of the WEJ is larger in the dawn side compared to the dusk side. We believe this is because the WEJ is in general better

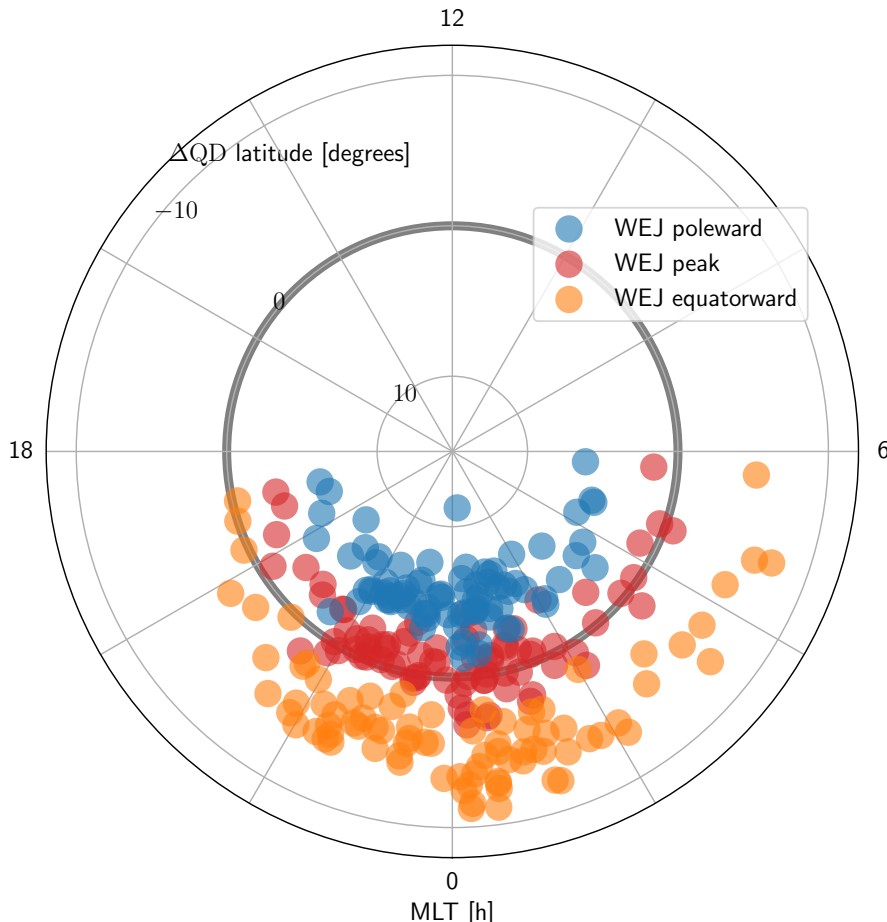

**Figure 10.** The MLT and $\Delta$QD latitude locations of the WEJ peaks and poleward and equatorward borders in the 2 hour MLT bin centered around the onset location 0 to 15 minutes after the onset. $\Delta$QD is the latitude relative to the onset latitude.

established in the dawn side. The Harang discontinuity could also be an explanation for the equatorward extent of the WEJ moving poleward in the dusk sector. To compare with the substorm time results we also checked 1797 oval crossings which
were within one hour west and one hour east of the SML value with any temporal distance to any substorm. For this check we also required that the SML MLT was between 18 and 6 hours. The latitudinal position distribution of the WEJ maximum was found to be centered around SML QD latitude. The median difference between the WEJ maximum and SML QD was 0.5 degrees poleward. However, unlike in the substorm time crossings shown in Fig. 10 there were several outliers far away from the SML QD latitude.

To study the amplitude evolution, we divided the data into pre-midnight onsets and post-midnight onsets. Of the 2976 onsets 1411 are located post-midnight and 1565 pre-midnight. Figure 11 shows the median WEJ data covering sectors W1 and W2 but now separately for the two sets of onsets covering the time period between -15 minutes before and 30 minutes after the onset

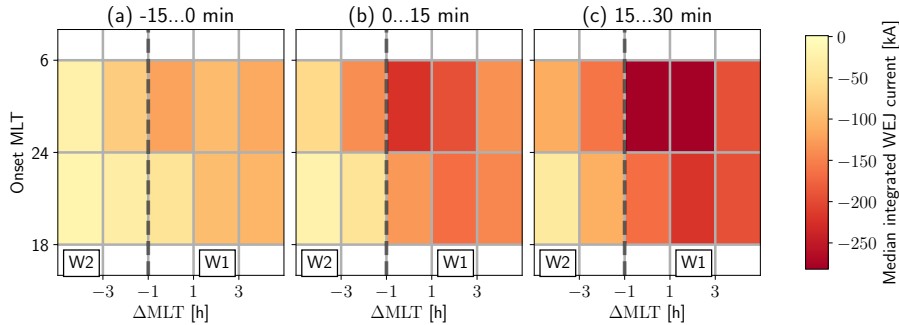

**Figure 11.** The time evolution of the median WEJ before and after the onset separated for pre-midnight onsets and post-midnight onsets. The top row shows the values for post-midnight onsets and the bottom row for the pre-midnight onsets.

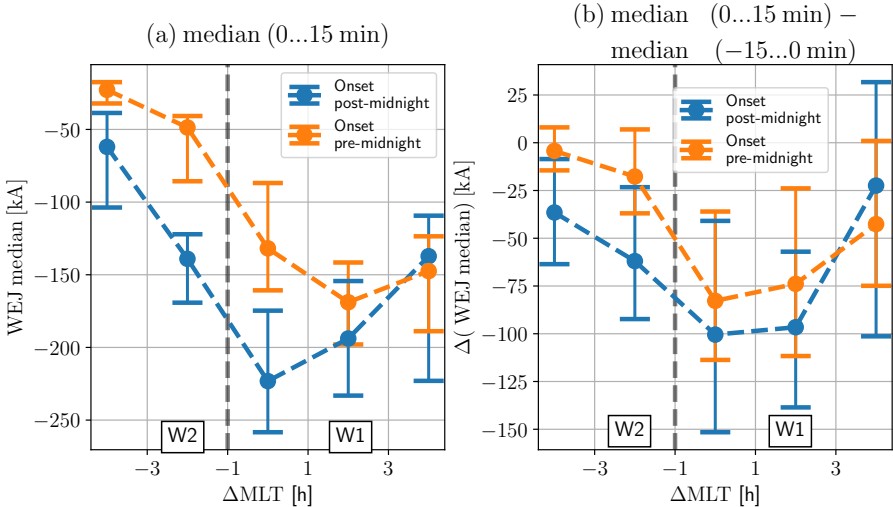

**Figure 12.** The median WEJ after the onset and the difference of the median WEJ between time steps before and after the onset separated for pre-midnight onsets and post-midnight onsets and bootstrapped 90% confidence intervals.

time. The three timesteps together contain 1260 oval crossings in the pre-midnight onset group and 1087 oval crossings in the post-midnight onset group. The basic statistic nature of the MLT distribution of the oval WEJ is underlying the changes in time,

as the pre-midnight values are clearly smaller and the pattern formed of the pre-midnight data is similar to the post-midnight data but shifted eastward. Figure 12 panel (a) shows the same data as Fig. 11 panel (b). The 90 % confidence intervals shown in the figure were calculated using the bias corrected and accelerated bootstrap method ($BC_a$) (Efron, 1993; Chernick, 2011). The pre-midnight curve has a minimum east of the onset sector. By contrast, the post-midnight curve has a minimum within the onset sector. Panel (b) in Fig. 12 shows the bootstrap estimated difference in the medians, which were again calculated

with the $BC_a$ method, between the last bin before and first bin after the onset, i.e. the difference of panels (b) and (a) in Fig.

11. The absolute value of the difference in the medians is greatest at the onset sector and the center of sector W1 for both the pre-midnight and the post-midnight curves with values of roughly 80 kA for pre-midnight onsets and 100 kA for post-midnight onsets. The pre-midnight values catch and take over the post-midnight values at the eastern edge of sector W1.

## 4 Discussion

Because of the local nature of the Swarm observations we must emphasize that our statistics consist of observations from different substorms at different locations in time and space and not from full-coverage observations of time evolving substorms. However, it's meaningful to interpret the statistics in relation to the physics of substorms through existing theories and compare the results with other studies. Following previous studies we will comment on the timing and expansion aspects of the data set. We concentrate on the WEJ because of its clearer relation to the SCW. We also note that the time scale and 1-D method used in

our analysis mean that our results consider mostly the large scale WEJ and can't give information of wedgelet type structures.

### 4.1 Temporal development of the amplitudes

To interpret the time development of our results we must note that as Gjerloev et al. (2007) point out it would be beneficial to use a normalized time scale in a similar fashion as they have done, in order to avoid mixing the expansion and recovery phases of substorms in a statistical study. For example Orr et al. (2019) also used the same normalized timescales. However, as the

Swarm oval crossings provide only snapshots of substorms from certain MLT sectors, it is not possible to avoid this mixing when working with Swarm data alone and we have to keep this in mind when looking at the MLT distribution of the currents at different times. Gjerloev et al. (2007) obtained approximately 30 minutes as the mean duration on the expansion phase and we can use this information to roughly assume that on average the bins from 0 to 15 and 15 to 30 minutes are mostly samples of the expansion phase of the substorms whereas the bins after the 30 minute mark are a mix of samples from expansion and

recovery phases and also recurrent substorm activity after the initial onset. This is supported by the large scale organisation of the temporal behaviour in Fig. 5 panels (b) and (c) which show general strengthening of the medians as well as the lower and upper quartiles. By contrast the medians are stable but ranges between quartiles are large from 30 minute mark onward in panels (d), (e) and (f) of Fig. 5 showing the mixing of different phases in observations.

Our observation of the median and the ranges of the WEJ reaching values close to their maximum values at 15 to 30 minutes

after the onset is consistent with the observations of Coxon et al. (2014) of maximum values for Region 1 and Region 2 FACs and their ratio. Forsyth et al. (2018) also observed similar timescales for the average Region 1 and Region 2 FACs to reach their maximum values. We conclude that the timescales for WEJ intensification coincide quite well with the FAC timing obtained from the global AMPERE observations.

### 4.2 WEJ amplitude and location in relation to the SCW and bulge type expansion

Figures 5 and 7 show that the enhancement of WEJ after the onset in sector W1 is located near or slightly equatorward of the onset latitude. In sector W2 the westward current is smaller in amplitude and located 2 to 4 degrees poleward. The MLT

sectors here are of course the edges of our bins and are not to be taken to be precise limits for any physical phenomena. In light of the SCW theory and observations of the expansion of bulge type substorms it is likely that the distribution in sector W2 is formed mostly of Swarm passes through the substorm bulge and the westward travelling surge. The W1 sector data on the other hand is formed of passes over the part of the SCW which flows along the auroral oval or eastern part of the bulge. Previous studies supporting this interpretation include for example Kamide and Akasofu (1975); Gjerloev and Hoffman (2002); Gjerloev et al. (2007); Fujii et al. (1994). We also note that the top currents in sector W1 and W2, especially in panel (d) in Fig. 5 are similar to values obtained by (Gjerloev and Hoffman, 2002) for the bulge and surge respectively in their model derived from Dynamics Explorer 2 satellite data (see Gjerloev and Hoffman (2002) Fig. 5). The latitudinal location of the observed WEJ in sectors W1 and W2 is qualitatively consistent with what could be expected from observations of satellite passes over a system depicted in Kepko et al. (2015) Fig. 9, which is based on observations of Fujii et al. (1994) and Gjerloev and Hoffman (2002). Gjerloev and Hoffman (2014) observed the similar displacement and amplitude differences of WEJ from SuperMAG data for the pre-midnight and post-midnight components of the electrojet determined from ground-based data of the SuperMAG network.

Figures 7 and 8 also provide a way to characterize the variability of the jet is in a statistical sense. We can interpret the non-overlapping ranges of the locations of the poleward boundary, the peak and the equatorward boundary as a sign of spatially and temporally stable jet in the chosen coordinate system and overlapping ranges as a sign of temporal or spatial variability, or both, in the system. Keeping this in mind we see not only that the WEJ is very clearly defined in the sector W1 throughout the studied period but the level of organization increases after the onset also in sector W2. The lower level of organization in the duskward sectors with overlapping locations and large variability in the amplitudes may arise from the satellite observing variable substructures or structures moving in time instead of a more statically positioned large scale jet as Kepko et al. (2015) anticipated.

The onset location is very well located inside the WEJ part of the oval and seems to always be quite close to the peak location which can be seen from clearly in Fig. 10. The distribution of the points shows the strong correlation of substorm onsets determined from the SuperMAG SML index and the WEJ profiles defined from Swarm data. This not surprising as the magnetic disturbances measured by the SuperMAG network should correspond to the ionospheric equivalent current which in turn should correspond quite well to the divergence free current derived from Swarm, but it is anyway an indication that the SML index based substorm detection does correlate with enhanced westward divergence free current with an electrojet-like profile centered on the location. Coxon et al. (2017) concluded that the substorm onset is co-located with the boundary of the Region 1 and Region 2 FACs which implies that the WEJ peak is also located close this boundary. Figure 11 shows that it is more likely to reach large currents if the onset location is in the post-midnight sector. This feature is the effect of the substorm enhancement being added to the pre-substorm WEJ which tends to be greater in the post-midnight sector. The actual median current value is clearly greatest at the onset location for the post-midnight onsets, while in the cases of pre-midnight onsets WEJ intensities tend to peak eastward of the onset. We note that the statistical observation of the WEJ peak location differing from the onset location for pre-midnight onsets arises very likely from mixing different substorms and pre-substorm conditions and does not mean that the SML index would not probe the maximum of the WEJ. By contrast, the difference in

the median before and after the onset shows the maximum enhancement occurring at the onset location for both pre-midnight and post-midnight onsets. It is likely that the differencing reduces the statistical effect of the underlying oval conditions and reveals better the substorm enhancement.

## 4.3   Limitations of the analysis and interpretation

Looking at Fig. 3 it is obvious that the number of oval crossings per bin is far from ideal for statistical analysis, ranging from about 40 to 125. The distribution of points is not very uniform across the bins. It is also possible that our quality flag selection allowing only cases where both EEJ and WEJ are entirely between the QD latitudes 50 and 85 causes systematic bias because certain current systems are not represented in the data set. We also recognize that estimating currents from single satellite magnetic field measurements with the SECS method involves solving an ill posed inverse problem. Although SECS has been shown to give reasonable results in statistical sense and in case studies (Juusola et al., 2007, 2016), some features in its output are affected by adjustments made in the inversion methodology for enhanced robustness in massive data analyses.

As mentioned in Sect. 3.2 the statistics show an enhancement of EEJ in sector E2 after the onset, which seems to propagate eastwards. The enhancement is located mostly poleward of the WEJ as can be seen from Fig. 8 and 7 and coincides also partly with the well defined WEJ sector. However, it's not clear if this a physical phenomenon or an artefact arising from the limiting 1-D approximation (ignorance of longitudinal gradients) used in single-satellite current products. In Fig. 1 we see the current profile with quite symmetrical eastward bumps on either side of the westward current. We also note that because the SML index is associated with westward flowing currents, the WEJ statistics are naturally more organised in our chosen coordinate system in general compared to the EEJ.

## 5   Conclusions

We have shown that the auroral electrojet characteristics derived from Swarm Swarm-AEBS data products is organized in a way which can be interpreted to be consistent with earlier observations of bulge-type substorm expansion and large scale SCW development. Although the data consists of separate oval crossings from different sectors of substorm current systems, the resulting distribution agrees with earlier studies of the time development of substorms at least in the 15 minute timescale used in this study. The peak currents are mostly observed between 30 and 45 minutes after the onset. The Swarm-AEBS data reproduces the well known poleward latitudinal displacement of the western part of the SCW in relation to the onset latitude and the eastern part of the SCW of about 2 to 4 degrees. Simultaneously we show the amplitude of the WEJ to be at least twice as large in the sector of 1 hour west to 1 hour east of the onset compared to values further than 1 hour west of the onset. The results also place the onset location determined by the SuperMAG method within the WEJ determined from Swarm so that the latitude of the onset in the SuperMAG database correlates well with the peak location of the WEJ determined from Swarm-AEBS data set regardless of the onset location. We also show that the $\Delta$MLT distribution of westward divergence free currents between 0 and 15 minutes after the onset is different for post-midnight and pre-midnight onsets most likely due to the variance caused by the underlying auroral oval conditions. However, the greatest temporal strengthening of the median WEJ

coincides with the SuperMAG onset location for both post-midnight and pre-midnight onsets. Our study shows that despite of
their different approaches SuperMAG and Swarm-AEBS data products can give a coherent picture of the main features in the
substorm current system. This finding encourages the combined usage of the two datasets in order to improve spatial coverage,
resolution and uncertainty estimates in comparison to results derived from single source data.

*Data availability.* The original and constantly updated data for the electrojet currents and boundaries (Kervalishvili et al., 2020) (Products
AEJxLPS_2F and AEJxPBS_2F) are available from https://swarm-diss.eo.esa.int/, last access: 22 June 2021. The Newell and Gjerloev
(2011a) substorm list is available from the SuperMAG website https://supermag.jhuapl.edu/substorms/, last access: 22 June 2021.

*Author contributions.* SK produced the manuscript with contributions from all co-authors. SK provided conceptualisation, investigation,
formal analysis and data visualization. AV and LJ were responsible for conceptualisation, data curation, software development, supervision,
funding acquisition. KK was responsible for funding acquisition, project management and supervision.

*Competing interests.* The authors declare that they have no conflict of interest.

*Acknowledgements.* This work has been funded by Academy of Finland (decision no. 314670). The authors thank the European Space
Agency for the magnetic field data and making Swarm data publicly available. We gratefully acknowledge the substorm timing list identified
by the Newell and Gjerloev technique (Newell and Gjerloev, 2011b), the SMU and SML indices (Newell and Gjerloev, 2011a), the Super-
MAG collaboration (Gjerloev, 2012) and SuperMAG collaborators
(https://supermag.jhuapl.edu/info/?page=acknowledgement, last access: 22 June 2021).

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
