# Peer review of "Spatio-temporal development of large scale auroral electrojet currents relative to substorm onsets"

_Annales Geophysicae, 2021_

## Author Comment (AC1)

We thank the anonymous referee for the helpful comments, and constructive remarks. Our replies are marked with blue color.

**Scientific comments**

Line 62–68: You mention CHAMP and Swarm here as examples of spacecraft orbiting above the current sheet that can identify the current systems, but mention of AMPERE (Anderson et al., 2000, 2002, 2014, 2018; Waters et al., 2001, 2004, 2020; Coxon, Milan and Anderson, 2018) might also be warranted in that context.

We will include a mention of AMPERE as it is indeed well warranted here.

Line 93–96: "only the largest areas in amplitude are defined as electrojets" – this is sensible, but the authors don't explain how this is done; an explanation of the selection criteria should be included to aid reproducibility.

We will include a small explanation of the method that is used in Swarm AEBS.

Line 104–109: The authors choose a separation of two hours as a method for interpreting the times between onsets as a "quieter baseline" than the times closer to the onsets. I have two comments here:

1. When discussing substorm recurrence and similar, the recurrence timescale is around 2.5 hours in a chain of substorms during sustained solar wind driving (Freeman and Morley, 2004). This means that using a criterion of under 2.5 hours means that periods which are during enhanced/sustained driving will be selected for analysis. This appears to be at odds with the desire for a quieter baseline, and is potentially a valid approach but which should be discussed here in the context of Freeman and Morley.

We plan to run the analysis again with the 2.5 hours limit and discuss this choice of limit in the context of Freeman and Morley, 2004. It is not expected that the results will change much.

2. The authors reference Forsyth et al. (2015) earlier in the manuscript and this dataset would provide a perfect way to disambiguate quiet interludes from sustained driving if the authors decide to do so. The SOPHIE technique described in that paper allows for expansion phases preceded by quiet times to be identified separately to expansion phases preceded by recovery phases, and if the authors decide to move away from two hour criterion in light of the potential contamination from periods of sustained driving, this would be a good method to capture the original motivation.

We agree that the SOPHIE technique would be capable of capturing quiet interludes. We believe that the 2- or 2.5-hour limit in the SuperMAG list is on its own enough to capture the intention to describe the statistical behavior of the electrojets after a quieter period. We think this is supported by the data as there is no clear evolution of the statistical values before the onset.

Line 132–133: While it may be true that the coordinate disparity is not the largest source of error, eliminating the sources of error which are within the control of the authors is a necessary step in conducting scientific analyses. As such, I would ask the authors to present the SuperMAG data in QD coordinates in their next submission.

We will redo the analysis and use QD coordinates also for the SuperMAG data.

Line 142–143: "...we observe the dawn and dusk electrojets dominating the lower right portion of the panel (a) and lower left portion of panel (b), i.e. the pre-onset parts of sectors W1 and E1 respectively. A decrease (i.e. a strengthening in amplitude) in the WEJ median after the onset is clearly visible in sectors W1 and W2." I might be misinterpreting what the authors mean here, but I don't see this. The dominant portions of the panels appear to be toward the upper halves, not toward the lower half.

The intent was to indicate the trace of the electrojets in the plots before the onset. We will reword the expression to be clearer.

Line 143–144: "A decrease (i.e. a strengthening in amplitude) in the WEJ median after the onset is clearly visible in sectors W1 and W2." I don't understand how the strengthening in amplitude of the median is a decrease in the median. It looks to me like the current in the electrojets increases after onset from the plotted figure, and I am confused by the authors' interpretation here. It also seems to be at odds with the next sentence, on lines 144–147, in which it says that the current increases after onset.

As we have retained the sign in the median integrated values and the WEJ value is always negative. Therefore, more negative values and decreasing median correspond to stronger electrojets. We agree that the meaning of increase of current is ambiguous in this context, and we will reword the sentence.

Line 147–149: "The most remarkable feature in panel (b) is the strengthening of the eastward current median values in sector E2 after the onset. The values are roughly doubled in this sector and the intensification seems to reach the maximum eastward extent only after 15...30 min after the onset." It seems to me that the disparity of the colour scale between before/after onset is similar

between E1 and E2, and so I'm not sure I agree that E2 is the most remarkable feature. I'd consider plotting these as percentage differences from the onset value (perhaps as Figures 4c and 4d) so that it's easier to compare the relative strengths pre- and post-onset.

We can plot the percentage differences and take a closer look.

Figure 5–6: Instead of plotting quantities in units of 10^5 A, you might want to plot them in kA because you spend a lot of time discussing the units in kA. Mentally converting back and forth between the text and figures makes it more difficult for the reader to follow. Additionally, I would recommend plotting the locations of W1 and W2/E1 and E2 on the axes, which would also make the text easier to follow.

We will change the unit to kA and plot the location of the sectors on the axis.

Figure 7–8: I found it difficult to interpret what these graphs were showing. I initially assumed that the north/south WEJ were referring to the WEJ in the Northern and Southern Hemispheres observed by Swarm, but after some thought, I instead assumed that WEJ peak must be the peak of the WEJ, and that the WEJ north/south must be the northmost and southmost reach of the electrojet in the Northern Hemisphere. It wasn't until seeing Figure 10 that I was confident of this interpretation. The north and south traces are not discussed in the text nor in the caption, but they do have some interesting implications for the shape of the electrojet and how that evolves over time, so I would recommend going into more detail on what this graph shows. I'd also recommend using "poleward" and "equatorward" instead of "north" and "south", since the former terms are much less likely to be misinterpreted.

We will change north and south to poleward and equatorward and add discussion of the plot within the limits of uncertainty provided by the SECS resolution.

Figure 10: Again, the larger spread on the dawn side than on the dusk side is interesting; it would be worth going into detail on this feature in the manuscript.

We agree with the referee that spreading in the dawn sector deserves some additional discussion in the paper. We believe the western jet has a larger extent in the north south direction in the dawn sector as the westward jet is naturally more established in this region. Our hypothesis is that signs of the Harang discontinuity can be seen in the dusk sector.

Line 195: Why use the 75% confidence interval? This seems low to me: is there a reason for this?

The confidence interval was chosen to present a coarse indicator of the uncertainty of the calculated values. We don't expect a larger value to change the interpretation, but we can choose a larger % confidence interval also.

Line 223–225: Coxon et al. (2017) looked further at the spatiotemporal development of substorms in AMPERE and found further evidence for the timescale here, but also found that the onset latitude was colocated with the R1/R2 field aligned current interface, which may be an interesting point of comparison to your finding that the peak WEJ coincides with the 0° line in QD coordinates (also relevant at lines 251–252).

This is an interesting point, and we will add discussion of this to the paper.

Line 231–234: To better make the link between W1, W2, and the substorm morphology described I would recommend including a schematic diagram which illustrates the proposed spacecraft passes and links them to Figure 5/7/9 to show how the results are what you would expect from passes through those currents.

We believe Figure 9 shows schematically the morphology and amplitudes. We also believe the addition of the sectors W1 and W2 in all the applicable figures will also illustrate this link.

Line 245–250: How do you differentiate between a well-defined large-scale jet which moves in time, and a set of variable substructures which are poorly defined but do not move in time? I would argue that the R1/R2 current systems are well-defined, but because they move in latitude with the expansion and contraction of the polar cap an initial reading of this passage makes me think they would be considered disorganised/badly defined, which seems incongruous to me.

This is true and we plan to adjust the paragraph to take the view into account.

Line 255–257: It would be good to compare randomly selected SML values and the westward DF current, i.e. to repeat the analysis without the substorm consideration. Naïvely, I would have expected the correspondence to be high at all times and not just during substorm times.

We agree with the referee and we can investigate this.

Lines 269–280: It would be nice to see discussion here of the fact that the EEJ is less well-organised in the paradigm you're discussing and why that is.

We can add further discussion to the manuscript. We believe the ultimate reason is that the SML probes the westward electrojet by nature and choosing

the origin of the coordinate system according to this index will thus lead to better organization of the WEJ data.

**Typographical errors**

In general in English style, "1...5" is not the style used to indicate a range; these should be replaced with "1–5" throughout.

We will check the indications of ranges in the text. While en dashes are mentioned in the English guidelines and house standards of the Copernicus manuscript preparation manual, there is also a point in the Mathematical notation section where it told that "A range of numbers should be specified as "a to b" or "a...b". The expression "a–b" is only acceptable in cases where no confusion with "a minus b" is possible."

Line 115: "close the poles" should be "close to the poles".

Figure 3: "(b) (corresponding to panel (a) in Fig. 4)" should be "(b) (corresponding to panel (b) in Fig. 4)".

Figure 8: "coveredby" in the caption needs a space.

Line 234: "over the part of SCW" should be "over the part of the SCW".

We will fix the typographical errors.

---

## Author Comment (AC2)

We thank the anonymous referee for the helpful comments, and constructive remarks. Our replies are marked with blue color.

**Specific comments**

*Introduction*

The introduction provides a good summary of the field and how this research fits into that. It could perhaps use a few more references in places to highlight as the authors say that it 'is still an active research topic' with many different ideas and opinions.

We agree and will add more references to studies highlighting the different ideas and opinions.

Line 35: A few more references to the recent interest in wedgelets may be useful eg. Plus an acknowledgement that their role is still very much up for debate.

We will expand the references relating to wedgelets and clarify the uncertainty of their exact role.

The amount of acronyms in the intro is very hard to follow! I would suggest the authors consider if all the acronyms are necessary, particularly two word acronyms such as AEJ, CF, DF etc. For example, given that 'curl-free' shouldn't even increase your word count and is only used three times I think using CF makes the manuscript harder to follow. AEJ is defined twice but the authors have still used 'auroral electrojet' several more times within the text. Please consider if all are necessary and if the authors decide they are, please check for consistency throughout the text.

This a good point and we will modify the usage of acronyms and pay attention to the consistency across the text.

Lines 55-63 would benefit from a few references.

We will take this into account.

*Data and Methods*

Line 89-90: This sentence is confusing. If it's pole is in Quasi-Dipole coordinates how is it Semi QD? I assume QD is quasi-dipole but this is not clear. Please restructure this sentence to make it clearer.

We will restructure the sentence. The Semi QD (quasi-dipole) is not equivalent to QD as the semi QD is a "normal" orthogonal spherical coordinate system where we can easily define an orthonormal basis. The QD basis is not orthonormal.

What years does your dataset include? How many events does this study include?

The years used have been mentioned in the text but will add also more information about the total amount of events. We will also correct the years included. The correct value should be 25 November 2013 – 31 December 2019.

Line 101: Could you explain that SML is an auroral electrojet index and a bit more about what it is and why it is used for the SuperMAG list? What are the benefits and negatives of defining onset purely from SML? Why has this list been used over other list such as the SOPHIE technique or the Frey list?

We will add more explanation of the SML and clarify the choice of the SuperMAG list. As the SuperMAG list is not dependent on visual data, its coverage will be better than visual lists, although the list is naturally dependent on the location and spacing of the contributing ground magnetometers. The Frey list does not overlap with Swarm lifetime so it cannot be used in this analysis.

Using the SOPHIE technique is technically possible, but the SuperMAG list was chosen because of the easy availability and accessibility of the list at the start of this study. It could be interesting to compare the analysis with different lists in future work, but we believe this is out of the scope of this paper.

Forsyth, C., et al. "A new technique for determining Substorm Onsets and Phases from Indices of the Electrojet (SOPHIE)." Journal of Geophysical Research: Space Physics 120.12 (2015): 10-592.

Frey, H. U., S. B. Mende, V. Angelopoulos, and E. F. Donovan (2004), Substorm onset observations by IMAGE-FUV, J. Geophys. Res., 109, A10304, doi:10.1029/2004JA010607.

The SM stands for superMAG as it is the SuperMAG AL index so it shouldn't be necessary to write 'SuperMAG SML'. Information and references are available in the indices section on the superMAG website.

We will modify the text accordingly.

Line 110: The example chosen for figure 1 and 2 is actually quite an extreme event. Could the authors comment on this and how it effects figure 1 and 2?

What would a smaller, more typical event look like? If one of the very high bins in later figures contained this or other extreme events would that cause significant inflation of the values?

The example is quite extreme, but we believe that it gives a good illustration of the probed parameters from the AEBS data. As we only calculate medians and percentiles, we do not expect these extreme events to affect the result very much. Certainly, the effect would be greater in means.

Line 113: Are you associating the time and MLT location for each auroral oval crossing with the substorm onset parameters? It would be helpful to reword this sentence slightly.

We identify the time and MLT and latitude for each auroral oval crossing.  The same parameters are provided for the onsets by the substorm list. The parameters for the oval crossings are then binned in relation to the nearest substorm fulfilling our qualification of temporal separation of the previous substorm. We will reword this to make the sentence clearer.

**Results**

Lines 141-143: What in the plot is supposed to show me that the dawn dusk electrojets are dominating? The slightly darker colours in the dawn dusk sector? Please explain how I am to interpret the plots.

Our interpretation is that the slightly darker colors before the onset are the indeed the result of the onset locations being located around the nightside, thus statistically positioning the westward jet towards positive MLT differences and the eastward towards negative MLT differences. This is perhaps more clearly seen in the (a) panels of Figures 5 and 6. Our intention was to indicate the dominance only in the times before the onset, not the after it. This characteristic is what is more clearly seen in figures 5 and 6.

Could you state what the average onset location is for reference?

We will add the average location of the onset to the text.

Line 145-150: Is there a comment on the higher values of the EEJ in the E1 section 50-100 minutes after the onset?

We have not focused on the EEJ but we believe this is the time regime of analysis is where the mixing of substorm phases is quite large. It is also possible that there is much statistical variation, as the percentile ranges are quite large as can

been in Figure 6. Another possibility could be variation caused by different driving conditions and saw tooth events.

Figure 5-8: Can you mark on the W1, W2 etc MLT lines? This would aid reading the results section by avoiding the need to flick back to figure 3.

We modify the images to include the limits.

Section 3.2: It would help if the units in the figures were the same as those used in the text e.g. either write the y axis in kA if you want to write 50-150 kA in line 156, or write the y axis as x10^5 A and 0.5-1.5 x10^5 A.

We will modify the units to kA.

Line 166: It took me some time to figure out what was meant by 'intensification seems to move eastward'. Perhaps the authors could make this clearer?

A clearer explanation will be added to the text.

Figure 11: Could the authors again add W1 and W2 boundaries to this plot and comment on the difference in magnitude between pre and post. A comment that pre-midnight is the bottom and post is the top would save thinking time! Is there much difference in coverage for pre and post midnight? How many onsets in each?

A clearer explanation of the plot will be added. We will also add the sectors and onset and point coverage of the post and pre midnight sets.

*Discussion & Conclusion*

Clear and representative of the work.

**Technical corrections**

Line 4 & 45 & 77: ESA is used repeatedly without introducing the acronym. European Space Agency is then used in line 77 without the acronym.

Line 45: Where does the S in AEBS come from?

Line 50 & 60 & 240: You have used ground-based throughout the rest of the manuscript.

Line 64: Switch word order to "can also provide observations. ."

Line 64 & 87: Use FAC

Line 70: You have already defined AEJs above.

Line 86: You have already defined and used SECS many times.

Line 88 & 90: If you're going to use DF throughout use it here.

107: 18-6 hrs.

123: The evolution of the parameters of interest ARE then inferred..

278: two 'or'

299: updated

309: The superMAG webpage has went a bit funny.

We will fix the technical errors.

---

## Author Response (AR1)

We have taken into consideration the comments by the referees and revised the manuscript. The point-by-point replies and a list of changes are included in this document.

**Point by point reply to the referee comments after the revision.**

**Comments by referee 1**

Line 62–68: You mention CHAMP and Swarm here as examples of spacecraft orbiting above the current sheet that can identify the current systems, but mention of AMPERE (Anderson et al., 2000, 2002, 2014, 2018; Waters et al., 2001, 2004, 2020; Coxon, Milan and Anderson, 2018) might also be warranted in that context.

We have included a mention of AMPERE as it is indeed well warranted here.

Line 93–96: "only the largest areas in amplitude are defined as electrojets" – this is sensible, but the authors don't explain how this is done; an explanation of the selection criteria should be included to aid reproducibility.

We have included a small explanation of the method that is used in Swarm AEBS.

Line 104–109: The authors choose a separation of two hours as a method for interpreting the times between onsets as a "quieter baseline" than the times closer to the onsets. I have two comments here:

1. When discussing substorm recurrence and similar, the recurrence timescale is around 2.5 hours in a chain of substorms during sustained solar wind driving (Freeman and Morley, 2004). This means that using a criterion of under 2.5 hours means that periods which are during enhanced/sustained driving will be selected for analysis. This appears to be at odds with the desire for a quieter baseline, and is potentially a valid approach but which should be discussed here in the context of Freeman and Morley.

We have rerun the analysis again with the 2.5 hours limit and discussed this choice of limit in the context of Freeman and Morley, 2004.

2. The authors reference Forsyth et al. (2015) earlier in the manuscript and this dataset would provide a perfect way to disambiguate quiet interludes from sustained driving if the authors decide to do so. The SOPHIE technique described in that paper allows for expansion phases preceded by quiet times to be identified separately to expansion phases preceded by recovery phases, and if the authors decide to move away from two hour criterion in light of the potential contamination from periods of sustained driving, this would be a good method to capture the original motivation.

We agree that the SOPHIE technique would be capable of capturing quiet interludes. We believe that the 2- or 2.5-hour limit in the SuperMAG list is on its own enough to capture the intention to describe the statistical behavior of the electrojets after a quieter period. We think this is supported by the data as there is no clear evolution of the statistical values before the onset.

Line 132–133: While it may be true that the coordinate disparity is not the largest source of error, eliminating the sources of error which are within the control of the authors is a necessary step in conducting scientific analyses. As such, I would ask the authors to present the SuperMAG data in QD coordinates in their next submission.

We have redone the analysis using QD coordinates also for the SuperMAG data.

Line 142–143: "...we observe the dawn and dusk electrojets dominating the lower right portion of the panel (a) and lower left portion of panel (b), i.e. the pre-onset parts of sectors W1 and E1 respectively. A decrease (i.e. a strengthening in amplitude) in the WEJ median after the onset is clearly visible in sectors W1 and W2." I might be misinterpreting what the authors mean here, but I don't see this. The dominant portions of the panels appear to be toward the upper halves, not toward the lower half.

The intent was to indicate the trace of the electrojets in the plots before the onset. We have reworded the expression to be clearer.

Line 143–144: "A decrease (i.e. a strengthening in amplitude) in the WEJ median after the onset is clearly visible in sectors W1 and W2." I don't understand how the strengthening in amplitude of the median is a decrease in the median. It looks to me like the current in the electrojets increases after onset from the plotted figure, and I am confused by the authors' interpretation here. It also seems to be at odds with the next sentence, on lines 144–147, in which it says that the current increases after onset.

As we have retained the sign in the median integrated values and the WEJ value is always negative. Therefore, more negative values and decreasing median correspond to stronger electrojets. We agree that the meaning of increase of current is ambiguous in this context, and we have added additional explanation.

Line 147–149: "The most remarkable feature in panel (b) is the strengthening of the eastward current median values in sector E2 after the onset. The values are roughly doubled in this sector and the intensification seems to reach the maximum eastward extent only after 15...30 min after the onset." It seems to me that the disparity of the colour scale between before/after onset is similar between E1 and E2, and so I'm not sure I agree

that E2 is the most remarkable feature. I'd consider plotting these as percentage differences from the onset value (perhaps as Figures 4c and 4d) so that it's easier to compare the relative strengths pre- and post-onset.

We have plotted the ratios and discussed the results in the manuscript.

Figure 5–6: Instead of plotting quantities in units of 10^5 A, you might want to plot them in kA because you spend a lot of time discussing the units in kA. Mentally converting back and forth between the text and figures makes it more difficult for the reader to follow. Additionally, I would recommend plotting the locations of W1 and W2/E1 and E2 on the axes, which would also make the text easier to follow.

We have changed the units to kA and plotted the location of the sectors.

Figure 7–8: I found it difficult to interpret what these graphs were showing. I initially assumed that the north/south WEJ were referring to the WEJ in the Northern and Southern Hemispheres observed by Swarm, but after some thought, I instead assumed that WEJ peak must be the peak of the WEJ, and that the WEJ north/south must be the northmost and southmost reach of the electrojet in the Northern Hemisphere. It wasn't until seeing Figure 10 that I was confident of this interpretation. The north and south traces are not discussed in the text nor in the caption, but they do have some interesting implications for the shape of the electrojet and how that evolves over time, so I would recommend going into more detail on what this graph shows. I'd also recommend using "poleward" and "equatorward" instead of "north" and "south", since the former terms are much less likely to be misinterpreted.

We will change north and south to poleward and equatorward and add discussion of the plot within the limits of uncertainty provided by the SECS resolution.

Figure 10: Again, the larger spread on the dawn side than on the dusk side is interesting; it would be worth going into detail on this feature in the manuscript.

We agree with the referee that spreading in the dawn sector deserves some additional discussion in the paper. We believe the western jet has a larger extent in the north south direction in the dawn sector as the westward jet is naturally more established in this region. Our hypothesis is that signs of the Harang discontinuity can be seen in the dusk side.

Line 195: Why use the 75% confidence interval? This seems low to me: is there a reason for this?

The confidence interval was chosen to present a coarse indicator of the uncertainty of the calculated values. We have opted for the 90 % interval in the data in the revised manuscript.

Line 223–225: Coxon et al. (2017) looked further at the spatiotemporal development of substorms in AMPERE and found further evidence for the timescale here, but also found that the onset latitude was colocated with the R1/R2 field aligned current interface, which may be an interesting point of comparison to your finding that the peak WEJ coincides with the 0° line in QD coordinates (also relevant at lines 251–252).

We have added discussion to the manuscript.

Line 231–234: To better make the link between W1, W2, and the substorm morphology described I would recommend including a schematic diagram which illustrates the proposed spacecraft passes and links them to Figure 5/7/9 to show how the results are what you would expect from passes through those currents.

We believe Figure 9 shows schematically the morphology and amplitudes. We also believe the addition of the sectors W1 and W2 in all the applicable figures will also illustrate this link.

Line 245–250: How do you differentiate between a well-defined large-scale jet which moves in time, and a set of variable substructures which are poorly defined but do not move in time? I would argue that the R1/R2 current systems are well-defined, but because they move in latitude with the expansion and contraction of the polar cap an initial reading of this passage makes me think they would be considered disorganised/badly defined, which seems incongruous to me.

We have taken this to account.

Line 255–257: It would be good to compare randomly selected SML values and the westward DF current, i.e. to repeat the analysis without the substorm consideration. Naïvely, I would have expected the correspondence to be high at all times and not just during substorm times.

We investigated this and the correspondence is good for all times. We have added a mention in the manuscript.

Lines 269–280: It would be nice to see discussion here of the fact that the EEJ is less well-organised in the paradigm you're discussing and why that is.

We believe the ultimate reason is that the SML probes the westward electrojet by nature and choosing the origin of the coordinate system according to this index will thus lead to better organization of the WEJ data.

**Typographical errors**

In general in English style, "1...5" is not the style used to indicate a range; these should be replaced with "1–5" throughout.

Line 115: "close the poles" should be "close to the poles".

Figure 3: "(b) (corresponding to panel (a) in Fig. 4)" should be "(b) (corresponding to panel (b) in Fig. 4)".

Figure 8: "coveredby" in the caption needs a space.

Line 234: "over the part of SCW" should be "over the part of the SCW".

We have fixed the typographical errors and modified the expressions for ranges of numbers while trying to keep them in accordance with Copernicus guidelines.

**Comments by Referee 2**

**Specific comments**

*Introduction*

The introduction provides a good summary of the field and how this research fits into that. It could perhaps use a few more references in places to highlight as the authors say that it 'is still an active research topic' with many different ideas and opinions.

We have taken this into account in the revision.

Line 35: A few more references to the recent interest in wedgelets may be useful eg. Plus an acknowledgement that their role is still very much up for debate.

We have taken this into account in the revision.

The amount of acronyms in the intro is very hard to follow! I would suggest the authors consider if all the acronyms are necessary, particularly two word acronyms such as AEJ, CF, DF etc. For example, given that 'curl-free' shouldn't even increase your word count and is only used three times I think using CF makes the manuscript harder to follow. AEJ is defined twice but the authors have still used 'auroral electrojet' several more times within the text. Please consider if all are necessary and if the authors decide they are, please check for consistency throughout the text.

We have modified the usage of acronyms and pay attention to the consistency across the text.

Lines 55-63 would benefit from a few references.

We have taken this into account in the revision.

*Data and Methods*

Line 89-90:  This sentence is confusing. If it's pole is in Quasi-Dipole coordinates how is it Semi QD? I assume QD is quasi-dipole but this is not clear. Please restructure this sentence to make it clearer.

The sentence has been made clearer. The Semi QD (quasi-dipole) is not equivalent to QD as the semi QD is a "normal" orthogonal spherical

coordinate system where we can easily define an orthonormal basis. The QD is not orthogonal, and the basis is not orthonormal.

What years does your dataset include? How many events does this study include?

The years used have been mentioned in the text and we have added also more information about the total amount of events. We have corrected the years included. The correct value should be 25 November 2013 – 31 December 2019.

Line 101: Could you explain that SML is an auroral electrojet index and a bit more about what it is and why it is used for the SuperMAG list? What are the benefits and negatives of defining onset purely from SML? Why has this list been used over other list such as the SOPHIE technique or the Frey list?

We have added more information about the SML. As the SuperMAG list is not dependent on visual data, its coverage will be better than visual lists, although the list is naturally dependent on the location and spacing of the contributing ground magnetometers. The Frey list does not overlap with Swarm lifetime so it cannot be used in this analysis. Using the SOPHIE technique is technically possible, but the SuperMAG list was chosen because of the easy availability and accessibility of the list at the start of this study. It could be interesting to compare the analysis with different lists in future work, but we believe this is out of the scope of this paper.

Forsyth, C., et al. "A new technique for determining Substorm Onsets and Phases from Indices of the Electrojet (SOPHIE)." Journal of Geophysical Research: Space Physics 120.12 (2015): 10-592.

Frey, H. U., S. B. Mende, V. Angelopoulos, and E. F. Donovan (2004), Substorm onset observations by IMAGE-FUV, J. Geophys. Res., 109, A10304, doi:10.1029/2004JA010607.

The SM stands for superMAG as it is the SuperMAG AL index so it shouldn't be necessary to write 'SuperMAG SML'. Information and references are available in the indices section on the superMAG website.

We have modified the text accordingly.

Line 110: The example chosen for figure 1 and 2 is actually quite an extreme event. Could the authors comment on this and how it effects figure 1 and 2? What would a smaller, more typical event look like? If one of the very high bins in later figures contained this or other extreme events would that cause significant inflation of the values?

The example is quite extreme, but we believe that it gives a good illustration of the probed parameters from the AEBS data. As we only calculate medians and percentiles, we do not expect these extreme events to affect the result very much. Certainly, the effect would be greater in means.

Line 113: Are you associating the time and MLT location for each auroral oval crossing with the substorm onset parameters? It would be helpful to reword this sentence slightly.

We identify the time and MLT and latitude for each auroral oval crossing. The same parameters are provided for the onsets by the substorm list. The parameters for the oval crossings are then binned in relation to the nearest substorm fulfilling our qualification of temporal separation of the previous substorm. We have tried to make the sentence a bit clearer.

**Results**

Lines 141-143: What in the plot is supposed to show me that the dawn dusk electrojets are dominating? The slightly darker colours in the dawn dusk sector? Please explain how I am to interpret the plots.

Our interpretation is that the slightly darker colors before the onset are the indeed the result of the onset locations being located around the nightside, thus statistically positioning the westward jet towards positive MLT differences and the eastward towards negative MLT differences. This is perhaps more clearly seen in the (a) panels of Figures 5 and 6. Our intention was to indicate the dominance only in the times before the onset, not the after it. This characteristic is what is more clearly seen in figures 5 and 6. We have modified the text to make our intention clearer.

Could you state what the average onset location is for reference?

We have added the average location of the onsets to the text.

Line 145-150: Is there a comment on the higher values of the EEJ in the E1 section 50-100 minutes after the onset?

We have not focused on the EEJ but we believe this is the time regime of analysis is where the mixing of substorm phases is quite large. It is also possible that there is much statistical variation, as the percentile ranges are quite large as can been in Figure 6. Another possibility could be variation caused by different driving conditions and saw tooth events.

Figure 5-8: Can you mark on the W1, W2 etc MLT lines? This would aid reading the results section by avoiding the need to flick back to figure 3.

We have modified the images to include the limits.

Section 3.2: It would help if the units in the figures were the same as those used in the text e.g. either write the y axis in kA if you want to write 50-150 kA in line 156, or write the y axis as x10^5 A and 0.5-1.5 x10^5 A.

We have modified the units to kA.

Line 166: It took me some time to figure out what was meant by 'intensification seems to move eastward'. Perhaps the authors could make this clearer?

We have taken this into account in the revision.

Figure 11: Could the authors again add W1 and W2 boundaries to this plot and comment on the difference in magnitude between pre and post. A comment that pre-midnight is the bottom and post is the top would save thinking time! Is there much difference in coverage for pre and post midnight? How many onsets in each?

We have added an explanation to the caption.  We have also provided statistics of the onsets and oval crossings in the text and added the sectors.

*Discussion & Conclusion*

Clear and representative of the work.

**Technical corrections**

Line 4 & 45 & 77: ESA is used repeatedly without introducing the acronym. European Space Agency is then used in line 77 without the acronym.

Line 45: Where does the S in AEBS come from?

Line 50 & 60 & 240: You have used ground-based throughout the rest of the manuscript.

Line 64: Switch word order to "can also provide observations. ."

Line 64 & 87: Use FAC

Line 70: You have already defined AEJs above.

Line 86: You have already defined and used SECS many times.

Line 88 & 90: If you're going to use DF throughout use it here.

107: 18-6 hrs.

123: The evolution of the parameters of interest ARE then inferred..

278: two 'or'

299: updated

309: The superMAG webpage has went a bit funny.

We have taken these into account in the revision.

**List of major changes in the revision:**

1. We have rerun the analysis using QD coordinates for SuperMAG data also.
2. All figures expect figure number 2 have been updated either because of data change caused by the change to fully using QD coordinates, or because of acronym usage. Also the modifications suggested by the referees have been taken into consideration.
3. Figure 4 has two new panels showing the ratio of the medians to the values before the onset.
4. Text has been updated when applicable according to the referee suggestions and to take into account the changes in figures and interpretation of the results.

All text changes are reflected in the track-changes file but only the new versions of the figures are shown.

---

## Author Response (AR2)

We thank the anonymous referee for the constructive remark.

Point-by-point reply:

Comment:

Line 105-106: "However, only the largest areas in amplitude are defined as electrojets in the context of this study. The selection is done by comparing the integrated current values of each identified sequence of positive and negative current density." I'd still like to see more detail here; as it stands I wouldn't be able to reproduce this result from the paper. This piece should include a short description of the actual algorithm employed to perform this identification, or, if that would make the manuscript too long, a reference to supplementary material that goes into more detail.

Reply:

We have added a more detailed explanation of the algorithm for identifying the electrojets from current density profiles.

List of changes:

- A clearer explanation of the elecrojet identification has been added.
- Changed the wording of the sentences talking about Figure 12 in section 3.3 to better reflect the figure and to be consistent with the conclusions in section 4.2
- Small typo and word consistency fixes
- Position of figures was changed to allow for better position relative to the relevant text and prevent the figures from appearing too late in the text.